Effect of orienteering experience on walking and running in the absence of vision and hearing

Machowska Weronika weronika.machowska@awf.wroc.pl wmachowska@gmail.com 1
Cych Piotr piotr.cych@awf.wroc.pl 1
Siemieński Adam 2
Migasiewicz Juliusz 1
1 Department of Sports Didactics, University School of Physical Education in Wrocław , Wrocław , Lower Silesia , Poland
2 Department of Biomechanics, University School of Physical Education in Wrocław , Wrocław , Lower Silesia , Poland
Boonstra Tjeerd
Electronic publication date: 2019 Sep 26
Publication date: 2019
Volume: 7
Electronic Location ID: e7736
Received 2019 Feb 24; Accepted 2019 Aug 25
Copyright: ©2019 Machowska et al.
Copyright year: 2019
Copyright holder: Machowska et al.
License: This is an open access article distributed under the terms of the Creative Commons Attribution License, which permits unrestricted use, distribution, reproduction and adaptation in any medium and for any purpose provided that it is properly attributed. For attribution, the original author(s), title, publication source (PeerJ) and either DOI or URL of the article must be cited.
License URL: https://creativecommons.org/licenses/by/4.0/

Keywords: Walking without vision, Running without vision, Spatial orientation, Orienteering, Foot orienteering

Funding: The authors received no funding for this work.

==============================
Purpose

This study aimed to examine differences between track and field (T&F) runners and foot-orienteers (Foot-O) in the walking and running tests in the absence of vision and hearing. We attempted to determine whether experienced foot orienteers show better ability to maintain the indicated direction compared to track and field runners.

Methods

This study examined 11 Foot-O and 11 T&F runners. The study consisted of an interview, a field experiment of walking and running in a straight line in the absence of vision and hearing, and coordination skills tests.

Results

Participants moved straight min. 20 m and max. 40 m during the walking test and min. 20 m and max. 125 m during the running test and then they moved around in a circle. Significant differences between groups were found for the distance covered by walking. Differences between sexes were documented for the distance covered by running and angular deviations. Relationship between lateralization and tendencies to veer were not found. Differences were observed between Foot-O and T&F groups in terms of coordination abilities.

Conclusions

Participants moved in circles irrespective of the type of movement and experience in practicing the sport. Orienteers may use information about their tendencies to turning more often left or right to correct it during their races in dense forests with limited visibility or during night orienteering competition.

Introduction

Movement is the biological need of every living organism. Countless species travel over shorter or longer distances in order to survive in constantly changing environmental conditions. The reasons for these migrations may vary, but most of them are to acquire food, new territory or to reproduce. The research on the mechanisms behind the ability to move and navigate led to the discovery that temporal and spatial orientation is very diverse in all living organisms. In order to navigate, living organisms use magnetic field, angular changes in relation to celestial bodies, ultrasounds, wind drifts, sky polarization patterns, gravitational receptors in hearing organs, ferromagnetic mineral magnetite produced in internal organs or specialized nerve cells in the brain acting as an “internal GPS system” (Shapiro, 2015). In order to move, humans use coordination skills (Raczek, Mynarski & Ljach, 2002), including temporal and spatial orientation, rhythmization of movements, balance control, kinaesthetic differentiation, quick motor response, coupled movements, high frequency of movements, and motor adjustment.

Human navigation has attracted the interest of researchers for years and has been analyzed many times, both based on field experiments and in the laboratory and virtual settings. The studies on mechanisms behind the human ability to navigate also have concerned tests conducted among blind and healthy people in the absence of vision (Bestaven, Guillaud & Cazalets, 2012; Paquet et al., 2007; Souman et al., 2009). Human navigation has been examined based on walking tests in the absence of vision and hearing, to a previously seen goal at distances ranging from four m to 60 m (Commins et al., 2013; Day & Goins, 1997; Ellard & Shaughnessy, 2003; Israël et al., 2013; Loomis et al., 1992; Rieser et al., 1990; Thomson, 1983; Uetake, 1992) along a previously prepared track (straight, round and triangular). Other experiments concerned the movement of study participants straight ahead, at a given distance or within a specified time (Bestaven, Guillaud & Cazalets, 2012; Kallie, Legge & Schrater, 2002; Maus & Seyfarth, 2014; Paquet et al., 2007; Pollatou et al., 2009; Souman et al., 2009). These studies have shown that a human can go straight ahead with minimal errors of the accuracy of reaching the destination or maintaining a straight line. When moving on distances ranging from two m to four m, humans can reproduce a specific distance in the absence of vision accurately, but longer distances tend to be underestimated (Israël et al., 2013). As argued by Thomson (1983) while navigating to the previously seen destination, human creates a kind of map of the surroundings used to cover the required distance and reach the goal. Research has shown that a person navigating their way in an open area for time and direction given can go straight for approx. 20 m (Loomis et al., 1992; Rieser et al., 1990; Souman et al., 2009). Then he or she begins to follow a circle with a diameter of 10 to 100 m (Maus & Seyfarth, 2014) or to 20 m (Souman et al., 2009) and eventually returns to the same point. Experiments in an enclosed room have shown that a human can go straight to max. 30 m, and then follows a circle between 19 m and 658 m in diameter, but in 80% of cases, this diameter is smaller than 300 m (Bestaven, Guillaud & Cazalets, 2012).

It was observed that women tend to turn in one particular direction during the walking tests (Bracha et al., 1987; Day & Goins, 1997) and are characterized by a higher average rotation rate compared to men. Bracha et al. (1987) suggested that this is caused by gender hemisphere lateralization but Nielsen et al. (2013) stated that small increases in lateralization with age were seen, but no differences in gender were observed, as confirmed by Agcaoglu et al. (2015). Furthermore, some researchers did not find a specific tendency to turn right or left during participants’ attempting to walk straight ahead in the absence of vision (Kallie, Legge & Schrater, 2002; Kallie, Schrater & Legge, 2007; Mohr et al., 2004; Rieser et al., 1990; Vuillerme, Nougier & Camicioli, 2002), or the turns were too small to be further analysed (Thomson, 1983). As claimed by Day & Goins (1997), the tendency for turning left or right can be considered an additional manifestation of lateralization in human motor activity. However, other researchers have demonstrated no effect of lateralization on the tendencies of the study participants for turning (Bestaven, Guillaud & Cazalets, 2012; Cratty, 1967; Lenoir et al., 2006). Some researchers have also emphasized the negative effect of fear of the dark, fear of being lost or unknown environment on walking speed (Bestaven, Guillaud & Cazalets, 2012; Souman et al., 2009). Practicing orienteering helps reduce these anxieties (fear of the dark, fear of being lost in unknown terrain). That is why they may be minimized by orienteering experience.

Orienteering is a sport in which athletes have to cover the distance from the start to the finish line in the shortest time possible, choosing their way of reaching the individual control points marked on a special map, with the start and finish locations. The route usually runs through mountainous, wooded and often uninhabited terrain. Orienteering is a sport combining physical exercise with mental effort, and requires maintaining balance and agility in order to overcome obstacles effectively. One of the elements of orienteering training is running in the specific direction, i.e., the run from one point to another in a straight line. Running in the specific direction can be performed using a compass, the so-called running to azimuth or on a hunch (using the natural terrain objects seen in front of or behind the runner) and by measuring distances with the number of steps (stepping). The effect of following a given direction accurately is most important when covering a section of a run in a thick forest, where it is challenging to maintain the direction of the run due to the limited visibility of the destination (e.g., a dense forest or night run). Furthermore, maintaining direction is one of the techniques used to find control points in a terrain with few reference points. The correct finding of control points has a positive effect on the final result of the competition.

No previous studies have attempted to verify whether orienteering experience may influence the ability to go straight in the absence of vision and hearing. This represents a unique feature of the present study. Moreover, it was challenging to test sighted people with running blind. The results of this study may have both theoretical and practical implications. They may be used to support the thesis that it is possible to program the human’s “internal GPS”. On the other hand, the results show that practising orienteering (moving in unknown terrains) may help sense the changes of direction and prevent them. Such conclusions could strengthen the thesis that orienteering should be practiced by children and adults to help them develop senses responsible for spatial orientation. Finally, the results of this study may demonstrate why practicing orienteering helps elite orienteers to be better than novice athletes, especially in maintaining direction when visibility is limited.

Souman et al. (2009) confirmed in their research that the phenomenon of people who follow a circular route after getting lost is not a myth. However, their results have inspired us to verify whether many years of experience in orienteering affects the ability to move correctly in a given direction in the absence of vision and hearing. Furthermore, we decided to verify whether the speed of movement would affect the results of the experiment. For this purpose, an innovative running test was conducted in an open area, in the absence of vision and hearing.

Materials & Methods

Study participants

The experiment examined 22 people aged 24.5 years on average (± 4.34 yr.), including 11 foot orienteers (Foot-O) aged 24.09 (± 4.78 yr.) and 11 track and field (T&F) runners aged 24.91 (± 4.04 yr.). Each group consisted of five women and six men living in the Lower Silesia region of Poland. The Foot-O group consisted of athletes including world and Europe champions with many years of experience (first-class athletes), while the T&F control group was selected from long-distance runners with performance level at distances of five km and 10 km equivalent of that for orienteering (three female second class long-distance runners). Study participants included: 21 right-legged people and one left-legged person and 18 right-handed people, three both-handed people, and one person declaring left-handedness.

Procedures

This study was approved by the university research ethics committee (Senate Committee on Ethics of Scientific Research of the University School of Physical Education in Wrocław, Ref: 6/2013). The research consisted of three parts: interviews, an experiment in the airport and a test of selected coordination skills. Due to their specific character, the examinations were conducted in the airport in Szymanów and the facilities of the University School of Physical Education in Wrocław, Poland.

The research was scheduled for autumn and spring, with atmospheric conditions favorable for the experiments. The experiment in the airport was conducted after sunset, in good and windless weather. Before starting the walk and run tests in the absence of vision and hearing, all participants were familiarized with the aim of the experiment and the experimental methodology. All subjects signed the participant consent form, which contained information about expected effects, possible benefits and risk for participants. Subjects were allowed to discontinue participation in the experiment at any time during the tests without giving a reason and were informed about this option before the beginning of the experiment. None of the participants decided to withdraw from the test. The method of data collection and research procedure were designed based on the research by Souman et al. (2009).

Interview

Each participant took part in an interview which concerned their age, athletic experience, the highest athlete class and lateralization of upper and lower limbs.

Experiments in the airport

The experiment in the airport consisted of two tests. Participants were asked to walk five times in a given direction within five minutes and run one two-minute section. During the walking tests, 5-minute duration was used because authors tried to reproduce the conditions of the experiment carried out by Souman et al. (2009), which was an inspiration for this research. During the running tests, 2-minute duration was used due to the high speed of movement of the participants, and, above all, the safety of people participating in the experiment. Subjects were blindfolded with a light-impermeable band and had their ears covered (earplugs and noise-cancelling headphones) to isolate them from any stimuli that would make it easier for them to find the right way. The face and neck of the participant were tightly covered with the shawl and hood of the jacket according to the recommendations by Cratty (1967) and Souman et al. (2009). The velocity of movement and distance covered by study participants were measured using a GPS device (Garmin, model 76S with a recording frequency of one Hz). The tests were carried out when the accuracy of the device indicated a minimum of 1.8 m. Angular deviation was obtained by checking the appropriate trigonometric formula that was calculated based on the angle between the vector (X2–X1, Y2–Y1) and the vector (X3–X2, Y3–Y2). X’s and Y’s are the co-ordinates of the subsequent positions of subjects. The average curvature [rad/m] of the trajectory was calculated by dividing the change of angular deviation by the distance covered. The inverse of thus calculated curvature is the radius of the circle-like trajectory.

The task of the researchers was to watch, supervise and care for the person taking part in the experiment. People participating in the experiment walked and run at a comfortable and natural speed adjusted to the abilities and sense of safety of the participants. The oral instruction was the same as in Souman et al. (2009) research. In the case of the person’s approaching an obstacle, a fence or a neighboring field, the supervisor stopped the participant and guided him or her to a different direction. The experiments were conducted after dusk.

Coordination tests of motor skills

Coordination tests of motor skills according to Raczek, Mynarski & Ljach (2002) were performed to verify if there are any links between the results of field experiments and the level of coordination skills: temporal and spatial orientation, rhythmization of movements, balance and kinaesthetic differentiation.

Spatial orientation ability was measured by the “Walk to the goal” test (Raczek, Mynarski & Ljach, 2002). The participant’s task was to cover a five m section excluding the sense of sight with a black, light tight scarf and reach the center (X) of a circle with a diameter of 1 m, drawn at the end of the route. The distance was measured in centimetres from the point between the participant’s feet to the center of the circle. The test was performed five times. The result of the test was “the average result of five trials in centimetres” (Raczek, Mynarski & Ljach, 2002, 157). Furthermore, in order to verify whether the deviation from the direction of the walk was affected by the distance covered, five “Walk to the goal” tests were performed in the absence of vision over a distance of 10 m. The test procedure was the same as in the case of the five m.

The rhythmization of movements ability was measured by the “Run at a given pace” test (Raczek, Mynarski & Ljach, 2002, 165–167). The test began with the measurement of the time of the test run at 30 m, then the participant had to overcome 30 m by inserting the feet into the 11 hoops. The result was the difference in times in both trials. Before starting the main test, each person performed a warm-up and a single test of both gears on both hoops and a distance of 30 m. The participant accelerated to the run from a gravel path away from the pitch of 10 m to perform the test at maximum speed. The track before each test was set in the same way according to the formula. All hoops were identical and had a diameter of 60 cm.

Balance ability was tested by the “Walk on the balance beam” test (Raczek, Mynarski & Ljach, 2002, 164). The participant’s task was to walk over the balance beam two meters long for 45 s back and forth. The examined person made three attempts, and the result was the average distance expressed in meters calculated from two best trials lasting 45 s or until the loss of balance. Subjects walked in sports shoes with a rubber sole on the wooden balance beam 10 cm wide.

The ability of kinaesthetic differentiation was measured by the “Standing long jump from place at 50% of maximum effort” test (Raczek, Mynarski & Ljach, 2002). The examined person repeated a standing long jump three times, each time trying to get the maximum result. Then the participants made a jump with closed eyes trying to reach half of their previous distance. Later the participants obtained feedback in the form of a difference of the result in relation to the maximum jumps. Then, the test with closed eyes was done twice. The result was “the percentage of error or accuracy in differentiation of the strength according to the formula: the difference between the pattern and the result obtained x 100/50% max result (pattern)” (Raczek, Mynarski & Ljach, 2002, 152–153).

These types of tests were chosen due to their reliability, repeatability, and the expected relationships between them and walking and running tests in the absence of vision and hearing carried out in the airport.

Statistical analyses

Statistical analysis was performed using the STATISTICA 13.1 PL software package (StatSoft Polska). All statistical tests were verified with a significance level set at α = 0.05. The results were considered statistically significant when p ≤ 0.05. Because of the size of the groups and skewness of distributions non-parametric tests were used for analysis. The Mann–Whitney U-test was used to compare differences between groups Foot-O and T&F in walking, running and coordination tests, whereas the Friedman test was used to analyze all tests results [ranks]. The relationships between the tests results in the given groups were tested by the ρ-Spearman’s rank correlation coefficient for quantitative variables (velocity of movement and angular deviations in walking and running tests). The chi-squared tests was used for qualitative variables such as the tendencies for turning and correlation between lateralization and the results of walking and running.

Results

Analysis of angular deviation, distance and speed for walking and running tests in the absence of vision and hearing

The analysis of the results showed that in all walking and running tests, the participants followed the set direction for about 20 m to 40 m during the walking test and about 20 m to 125 m during the running test, and then they began to turn and move in circles. The participant who was observed to have accumulated the angle of at least equal to π [rad] was adopted as the person moving in circles. In other words, the participant was starting to walk or run around if there were changes in the rate of increasing the angle from the value close to zero to a constantly increasing value in the plot of the dependence of the accumulated angle on the distance traveled. The results of the running test for all the subjects are shown in Fig. 1. All test results were transformed to start from one starting point with coordinates (0.0) to the East. For this reason, the axes in the coordinate system should be interpreted as corresponding to cardinal directions: East-West (X axis), North-South (Y axis).

Figure 1 Results of running tests in the absence of vision and hearing.

(T&F-W1) The first women from track and field group. (T&F-W5a and T&F-W5b) The trail divided into two parts a and b, because the fifth woman from track and field group has been stopped before the obstacle. (Foot-O-W1) The first women from foot-orienteering group. (T&F-M1) The first men from track and field group. (Foot-O-M1) The first men from foot-orienteering group.

There were no significant differences between Foot-O and T&F groups in angular deviations in walking (p = 0.896) and running (p = 0.264) tests in the absence of vision and hearing (Fig. 2). No correlations were observed between the results of walking and running tests in angular deviations for the entire group (p = 0.790), and the differences were statistically significant. Comparison of all results of walking and running tests for both groups revealed that the difference was significant (U = 132.00; Z = 2.57; p = 0.010). It means that higher angular deviation was observed during walking compared to running. Examination of angular deviations between walking and running separately for Foot-O and for T&F groups showed no differences in angular deviation in the Foot-O group (p = 0.264), but they were present in the T&F group (U = 24.00; Z = 2.36; p = 0.018).

Figure 2 Results of mean angular deviations from a given direction of movement during walking and running tests in the absence of vision and hearing related to sport.

(Foot-O) Foot-orienteering group. (T&F) Track and field group.

Significant differences in angular deviations were observed between women (Me = 0.039) and men (Me = 0.018) during walking (U = 27.00; Z = 2.14; p = 0.032) but they were not observed during running (p = 0.489). Women changed the set direction of movement to a greater degree and reached higher mean value of angular deviations (Fig. 3), i.e., they turned more from the given direction of movement, and thus followed the circular path more often.

Figure 3 Results of mean angular deviations from a given direction of movement during walking and running tests in the absence of vision and hearing related to participant sex.

Compared to group Foot-O (Me = 348.6), a significantly longer total distance (U = 30.00; Z =  − 1.97; p = 0.049) during the walking tests was covered by group T&F (Me = 402.2) (Fig. 4). During the running tests, the only significant differences were observed between sexes, with men (Me = 386) covering a significantly longer distance (U = 19.00; Z =  − 2.67; p = 0.008) than women (Me = 272).

Figure 4 Distance covered during walking and running tests in the airport in the absence of vision and hearing related to sport.

(Foot-O) Foot-orienteering group. (T&F) Track and field group.

Mean speed during walking in the airport in the absence of vision and hearing was 1.35 m/s (± 0.18 m/s), whereas the mean running speed was 2.50 m/s (± 0.58 m/s). No significant differences were observed in both walking and running tests, whereas the analysis of mean movement speeds revealed significant correlations in group T&F (ρ = 0.67; p = 0.023), in group of men (ρ = 0.63; p = 0.028). No difference in walking (p = 0.249) and running speed between men and women was observed (p = 0.223).

Comparison of walking and running tests in the absence of vision and hearing with coordination tests

Analysis of ranks (the Friedman test) of all tests conducted in the airport (walking and running in the absence of vision and hearing) and results of the coordination tests conducted for groups Foot-O and T&F did not show any significant differences between these groups. The group Foot-O turned out to be significantly better than group T&F in terms of coordination abilities (Table 1).

Table 1 Analysis of differences between groups (Foot-O and T&F) in tests (walking, running and coordination).

(Foot-O) Foot-orienteering group. (T&F) Track and field group. (U) The Mann-Whitney U test result. (Z) The Z test result. (p) Statistical significance.

Differences between groups [rank]	Tests	
	Walking	Running	Coordination	
	Mean	SD	Mean	SD	Mean	SD	
Foot-O	5.72	2.83	6.82	3.06	4.27	2.87	
T&F	6.36	3.85	5.27	3.52	7.82	2.76	
U	54.50	43.00	21.50	
Z	−0.36	1.12	−2.54	
p	0.717b	0.263b	0.011a	
Notes.

a Significant at the 0.05 probability level .

b Nonsignificant.

Furthermore, the analysis showed that the ranks of tests performed in the airport and the average rank of the tests of coordination abilities for the examined groups were consistent, i.e., no significant differences were found between the obtained results (Table 2) but on the other hand, the results of Spearman’s rank correlation test between coordination ability and walking (p = 0.672) and between coordination ability and running (p = 0.691) were insignificant.

Table 2 Comparison of the results (ranks) of tests in the absence of vision and hearing with coordination tests within groups (Foot-O and T&F).

(Foot-O) Foot-orienteering group. (T&F) Track and field group. (SD) Standard deviation. (χ2) The chi-squared test result. (p) Statistical significance.

Comparison of the results within groups [rank]	Tests	
	Walking	Running	Coordination				
	Mean	SD	Mean	SD	Mean	SD		χ2	p	
Foot-O	5.72	2.83	6.82	3.06	4.27	2.87		3.526	0.172a	
T&F	6.36	3.85	5.27	3.52	7.82	2.76		0.905	0.636a	
Women	5.50	3.03	5.50	3.03	5.50	2.98		0.229	0.892a	
Men	6.50	3.61	6.50	3.61	6.50	3.59		1.111	0.574a	
Notes.

a Nonsignificant.

Relationships between tendencies for deviating from a given direction in different walking and running tests in the absence of vision and hearing

In terms of the tendency for deviating from a given direction, it was observed that in walking and running tests in the airport, the respondents usually turned to the same direction (side). Some of them turned right (50%) or some left (45.45%), and one person (4.55%) had no preferred direction of turning. There was no difference between tendency to turn right or left between walking trails in the airport (p = 0.479), but no correlation was observed between walking and running in the airport (p = 0.402). The results collected from walking tests on the 5 m and 10 m distance in the absence of vision and hearing of the same athletes (one of the coordination task) showed the tendency to turn left (49.59% of all subjects), right (43.05%) and going straight ahead (7.36%). Again no correlation was observed between tendencies to turning to the same direction (side) on these two different distances (p = 0.451).

Relationships between lateralization and the results of walking and running tests in the absence of vision and hearing

Limb lateralization determined based on interviews shows that study participants included 21 (95.45%) right-legged people and one left-legged person and 18 (81.82%) right-handed people, three both-handed people, and one person declaring left-handedness. Analysis of the data (chi-squared maximal likelihood test) on lateralization of the upper and lower limbs in the participants and their tendencies for deviating from a direction of walking or running revealed no significant correlations between the variables studied (Table 3).

Table 3 Tendencies for deviating from a given direction in the absence of vision and hearing (walking and running) in the context of lateralization.

(χ2) The chi-squared test result. (p) Statistical significance.

Differences between lateralization and turning tendency	Tests	
	Walking	Running	
	χ2	p	χ2	p	
Upper limb	1.835	0.766a	2.149	0.708a	
Lower limb	3.132	0.209a	1.434	0.488a	
Notes.

a Nonsignificant.

Discussion

Scientists try to determine the reasons of moving into circles instead of going straight. Many experiments have confirmed the tendency of people for deviating from the route. Uetake (1992) claimed that these gait asymmetries are caused by a small structural or functional imbalance of the limbs. Some researchers found the effect of the asymmetry on walking in a circle (Maus & Seyfarth, 2014) while others have not shown such correlations (Bestaven, Guillaud & Cazalets, 2012; Cratty, 1967; Souman et al., 2009). The tendency to deviate from a given direction of movement may be influenced by the way of shifting the center of gravity, since the study participants who turned left during the walking tests transferred the body weight to the left, whereas those who transferred the body weight to the right turned right (Bestaven, Guillaud & Cazalets, 2012). Souman et al. (2009) argued that deviations from the straight-ahead direction are caused by the accumulation of sensorimotor system disturbances.

This study attempted to examine differences between groups Foot-O and T&F in walking and running tests in the absence of vision and hearing to check if orienteering training may improve ability to keep direction and prevent people from turning left or right. Surprisingly, there were no differences in angular deviations between these two compared groups. What could be reason of such results? The authors do not know exactly why orienteering experience did not help maintain the direction but it is possible to propose some hypothesis. The first one comes from the statement that people need some visual or auditory feedback to compensate postural or limb strength asymmetry which leads to deviation. The other hypothesis is that women who practiced orienteering were less skilled than women in the T&F group because of poorer selection and this influenced the results.

The T&F group turned out to be also significantly better in terms of walking distance in the absence of vision and hearing, whereas no significant differences were observed in the running test. This result may seem unexpected in light of the experiences gained by Foot-O athletes from participating in competitions and night orienteering training since all studies to date have shown the importance of experience and its impact on the speed of walking and running in cross-country settings (Ceugniet, 1991). Worse results achieved by orienteers (compared to runners) in walking distance were again substantially affected by significant differences in sexes (women moved much slower than male orienteers). Distance covered might be influenced by instruction given to participants about choosing a comfortable speed during the experiment because speed and distance were not factors with high significance.

Significant differences in the distance travelled between men and women seem unsurprising as they are related to differences resulting from sexual dimorphism. The level of anxiety may have influenced the speed of movement due to moving in the absence of vision and hearing but it needs further research. The mean walking speed (1.35 m/s) was similar to the mean speed (1.32 m/s) observed by Bestaven, Guillaud & Cazalets (2012), which suggests that sport’s experience does not help subjects move faster than average people do when they are deprived of hearing and sight.

Furthermore, the comparison between women and men showed that the direction of movement changed in women group to a greater degree. Women deviated more often from the straight line and usually followed a circle with a smaller radius than men. These results are consistent with the findings reported by other scientists, who demonstrated better abilities to navigate and assess the direction in men (Grön et al., 2000) and more frequent deviating from the direction in women in daylight (Bracha et al., 1987). It is difficult to state what causes that women have higher tendencies to deviation. Maybe it is caused by their higher body asymmetries and structural or functional imbalance of the limbs or maybe by some psychological factors but it needs further research.

Participants (both groups of athletes) achieved comparable results in mean value of angular deviation (respectively during the walking (0.026 rad/m) and running (0.017 rad/m) tests to those documented by Cratty (1967), who found that the participants turned on average by 0.021 rad/m during walking. In the test of running to a given direction in the absence of vision and hearing, no differences in angular deviations were observed in group T&F compared to group Foot-O but there were statistical differences within T&F group between walking and running. This could be caused by their running experience. It was also observed that study participants could walk in a set direction over a distance of about 20 m to about 40 m, and then they turned right or left and started to walk in a circle. Therefore, the results are consistent with those obtained by other researchers (Bestaven, Guillaud & Cazalets, 2012; Loomis et al., 1992; Maus & Seyfarth, 2014; Rieser et al., 1990; Souman et al., 2009). Much better results were obtained in straight line since it was observed in the running test that the participants covered as much as 125 m while maintaining the given direction. Furthermore, it was found that during walking, the study participants followed around a circle with a diameter of 8.7 to 25 m, whereas during running, this was from 90.91 m to 400 m. The results of the walking tests were similar to those documented by other researchers (Bestaven, Guillaud & Cazalets, 2012; Maus & Seyfarth, 2014; Souman et al., 2009), with the diameters of the circle in which the participants moved during the running tests larger than those reported by Maus & Seyfarth (2014) and comparable with those presented by Bestaven, Guillaud & Cazalets (2012). The results obtained in the study confirm the tendency of people to follow a circle but they show that during running, study participants can cover a longer distance in a given direction compared to walking. Our results are opposite to those published by Maus & Seyfarth (2014) who found that a shorter step during walking reduces the tendency for turning, whereas any change in speed leads to a change in the direction of the turn. It is likely that people who practice running are more confident when they run than when they walk in the absence of vision and hearing.

Humans show tendencies for deviating from a given direction of movement, both to the left and to the right (Bestaven, Guillaud & Cazalets, 2012; Cratty, 1967; Kallie, Legge & Schrater, 2002; Lenoir et al., 2006; Maus & Seyfarth, 2014; Pollatou et al., 2009; Rieser et al., 1990; Shapiro, 2015; Souman et al., 2009; Thomson, 1983; Uetake, 1992; Vuillerme, Nougier & Camicioli, 2002). Furthermore, Lenoir et al. (2006) showed that the tendencies for turning left while running are more significant than when walking (tests carried out with vision). Tendencies to deviate from a given direction of movement were also observed in tests of walking and running in a given direction in the airport area and during coordination tests of five m and 10 m ”Walk to the goal”. The analysis of the results of the above tests did not show any correlations in terms of the tendency for deviating from a given direction. However, the results of the analyses are consistent with the results documented by previous researchers (Vuillerme, Nougier & Camicioli, 2002), who also did not find a specific tendency to turn right or left while walking in a given direction of movement in the absence of vision. The results are also consistent with the observation of Paquet et al. (2007) who found the lack of repeatability in the aspect of the tendency for turning from a given direction of movement during walking tests in the absence of vision.

Specific coordination tests were performed to eliminate the influence of coordination abilities on results of walking and running in the absence of vision and hearing. It happened that orienteers (both men and women) are better coordinated than T&F athletes. The biggest difference was observed for the balance beam test. It was probably caused by better ability of orienteers to maintain balance. They practice balance during many training sessions and events held in natural settings (running on uneven ground, walking on fallen trees through the streams and rivers, moving on the rock edges etc.). Results of statistical tests (Friedman’s and Spearman’s) showed that there were neither dependence between coordination abilities and level of angular deviations in walking and running nor significant differences. Finally, it can be stated that the different level of participants’ coordination of Foot-O and T&F groups did not influence the results of the experiment.

The results collected from walking and running tests in the airport in the absence of vision and hearing concerning the tendency to turns are inconclusive. It can be observed that more than 50% athletes have tendencies to always turn to the same direction (12 people) in walking trails. They turned five times to the same side (left or right). So the tendency to turning in one specific direction is probably highly individual and maybe connected with some asymmetry. These results are similar to those obtained by Bestaven, Guillaud & Cazalets (2012) for walking tests, with 50% of the tests ending on the left, 39% on the right and 11% in a straight line. Lenoir et al. (2006) observed that 43.9% of the walking tests ended on the left, 41.1% were neutral and 15% ended on the right, while in the running tests, 68.2% of the tests ended on the left, 13.1% were neutral, and 18.7% ended on the right. Similarly, no correlation was found between tendencies to turning in walking and running and no correlation between tendencies to turning between walking in the airport and walking on the distance five m and 10 m in the coordination task.

Right-handed and right-legged individuals were prevalent in participants in our study and in studies published by other researchers (Bestaven, Guillaud & Cazalets, 2012; Day & Goins, 1997; Lenoir et al., 2006). Analysis of the data on lateralization of the upper and lower limbs of the study participants and their tendencies for turning from a given direction revealed no significant correlations between the variables. Therefore, the results obtained are consistent with those obtained by other researchers (Bestaven, Guillaud & Cazalets, 2012; Cratty, 1967; Lenoir et al., 2006).

Based on the results of the walking tests, no tendencies were observed for the right-handed and right-legged participants to turn from a given direction. Right-handed and right-legged persons turned left or right during the running tests, which is consistent with the findings of Day & Goins (1997). The results contradict the reports that right-handed people turn left whereas left-handed people turn right (Bracha et al., 1987) or the fact that during running, more frequent turning direction is to the left compared to walking (Lenoir et al., 2006).

The effects of sport-specific orienteering training on temporal—spatial orientation can be observed in terms of estimating the distance to the goal more accurate mapping of the road, and smaller angular deviations from the direction of walking in a given direction. A detailed analysis of the results of the tendencies for turning was presented to orienteering runners in order to eliminate errors when running in a given direction in the terrain with difficult visibility and runnability (vegetation density and type of ground affecting the running speed). This may help runners cover the route faster and find the subsequent control points more efficiently.

It is interesting how the human “internal GPS” can be programmed and how the effects of such training can be measured. These findings showed that walking and running tests in the absence of vision and hearing are not reliable and the authors must look for other solution to this problem.

Conclusions

Significant differences between sexes were observed for angular deviations in the walking test in a given direction in the absence of vision and hearing, distance covered and the tendency for turning to a specific direction in the test of “blindfolded” running. No correlations were observed between limb lateralization and the results of the field experiment. The study showed that the tendency for following a circular path observed in people who got lost also applies to experts in navigation such as orienteering runners when they are deprived of vision and hearing. Although orienteering athletes devote much time to training to maintain a given direction (also in the terrain with limited visibility and at night), no significant differences were found between group Foot-O and group T&F in terms of angular deviations during the walking tests and during the running test in the airport in the absence of vision and hearing. However, an analysis of the results concerning the tendencies for turning can help eliminate deviation from the direction in densely forested and challenging terrain. The present study has some limitations that should be taken into account. Firstly, the analysis of the results may be limited by a small number of respondents. Consequently, verification using the chi-squared (maximal likelihood) test led to the difficulty of rejecting the zero hypothesis and thus demonstrated a statistical significance of the observed phenomena. Secondly, the sports skill level of the examined athletes was diverse, which to some extent could have affected the results.

Further research on the level of abilities to maintain balance in orienteers should be conducted in the future since the results of the experiment presented in this study show that orienteers have an advantage over long-distance runners in the level of coordination abilities.

Supplemental Information

Dataset S1 Raw data with full explanation of data processing - one participant example

Click here for additional data file.

Dataset S2 Results of running test—running distance and accumulated angle

Click here for additional data file.

Dataset S3 Results of mean angular deviations from a given direction of movement during walking in the absence of vision and hearing

(Foot-O) Foot-orienteering group. (T&F) Track and field group. (W) Women. (M) Men.

Click here for additional data file.

Dataset S4 Results of mean angular deviations from a given direction of movement during running test in the absence of vision and hearing

(Foot-O) Foot-orienteering group. (T&F) Track and field group. (W) Women. (M) Men.

Click here for additional data file.

Dataset S5 Results of mean covered distance during walking tests in the absence of vision and hearing

(Foot-O) Foot-orienteering group. (T&F) Track and field group. (W) Women. (M) Men.

Click here for additional data file.

Dataset S6 Results of mean speed during walking tests in the absence of vision and hearing

(Foot-O) Foot-orienteering group. (T&F) Track and field group. (W) Women. (M) Men.

Click here for additional data file.

Dataset S7 Results of mean distance covered during running test in the absence of vision and hearing

(Foot-O) Foot-orienteering group. (T&F) Track and field group. (W) Women. (M) Men.

Click here for additional data file.

Dataset S8 Results of mean speed during running test in the absence of vision and hearing

(Foot-O) Foot-orienteering group. (T&F) Track and field group. (W) Women. (M) Men.

Click here for additional data file.

Dataset S9 Results and ranks of all tests

(Foot-O-W1) The first woman from foot-orienteering group. (T&F-W1) The first woman from track and field group. (Foot-O-M1) The first man from foot-orienteering group. (T&F-M1) The first man from track and field group.

Click here for additional data file.

Dataset S10 All participants ranks in walking, running and coordination tests

(Foot-O) Foot-orienteering group. (T&F) Track and field group. (W) Women. (M) Men.

Click here for additional data file.

Dataset S11 Results of lateralization and turning tendency in all tests

(Foot-O) Foot-orienteering group. (T&F) Track and field group. (W) Women. (M) Men. (−1) Right. (0) Right and left. (1) Left.

Click here for additional data file.

The authors would like to thank all participants for their help and involvement.

Additional Information and Declarations

Competing Interests

Author Contributions

Human Ethics

Data Availability

The authors declare there are no competing interests.

Weronika Machowska conceived and designed the experiments, performed the experiments, analyzed the data, contributed reagents/materials/analysis tools, prepared figures and/or tables, approved the final draft.

Piotr Cych conceived and designed the experiments, performed the experiments, analyzed the data, prepared figures and/or tables, authored or reviewed drafts of the paper, approved the final draft.

Adam Siemieński analyzed the data, prepared figures and/or tables, approved the final draft.

Juliusz Migasiewicz authored or reviewed drafts of the paper, approved the final draft.

The following information was supplied relating to ethical approvals (i.e., approving body and any reference numbers):

The University School of Physical Education in Wrocław, granted Ethical approval to carry out the study within its facilities (Ethical Application Ref: 6/2013).

The following information was supplied regarding data availability:

The raw measurements are available in the Supplemental Files.

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
