# Peer review of "Effect of orienteering experience on walking and running in the absence of vision and hearing"

_PeerJ, doi:10.7717/peerj.7736_

## Round 0.1 · original submission · Major Revisions

The reviewers raised several questions regarding the statistical analysis and interpretation of the results. In particular, the authors should justify the sample size used for different statistical analyses and interpret the results in light of the statistical power. Moreover, the authors should provide a more in-depth interpretation of the results in the Discussion.

·

Basic reporting

No raw data and data processing (i.e. codes) have been provided in the Supplementary files, I wonder if it would be possible to provide raw data, codes, and documentation that makes interpreting these data (or performing future analyses) easier for readers.

Experimental design

Introduction
How the results of this study add more insight in the field and how the results can be used in practical? Would be possible to clearly mention this in Introduction that make this paper more interesting.
Material and Methods
Line 155:I didn’t understand why 5 minutes and 2 minutes were used for walking and running tests, respectively.
Line 164-166: This kind of instruction influences the calculated variables (i.e. covered distance, angular orientation, and speed). The space should be big enough and without any obstacle, but apparently it was not.
I didn’t understand how and why the coordination tests have been performed and how the data has been obtained from these tests. The coordination tests and related data processing need to be fully described.
Statistical analyses
It was unclear to me exactly which statistical tests were used to compare walking and running, men and women, Foot_o and T& F groups, and to correlate between lateralization and the results of walking and running. I would suggest you to explicitly mention which statistical method has been used for which.
Results
Line 199-205: No p-values have been reported here.
Too much information has been shown in Table 1, 2,3, and 4. It is hard to reader to understand these results. I would suggest you to show these results in figures, rather than tables.
Discussion
It has just been mentioned that the results are consistent or contradict with previous studies and the paper has been stopped in this point. For example in Line 349, it has been mentioned that the results are contradict with Bracha et al.. Why the results are contradict? Not only it is important to mention consistency and contradictory, but also it is important to mention that why and how the results are consistent and contradict.
In the Discussion section the authors mostly described results. The usefulness of the findings was missing.
Lines 281-282: It is important to know how you ended up with this statement. Which task you mean? Based on which results Foot-O had much greater angular deviation than T &F? Please clearly mention the task and refer this statement to the related results that has been presented previously and follow this through the whole discussion. After showing the results in figures, I would suggest you to explicitly draw a link between Discussion section to Results section.
Line 301: How the diameter has been calculated? Please clearly mention this in Method section.

Validity of the findings

Too many statistical tests have been done without correction. In some cases the statistical test has been performed on very small sample size, such as comparison between men and women (N=5). Therefore, these statistical tests are most likely underpowered and the probability of type I error with such a small sample size is very high. I would suggest you to perform correction tests and take out all the statistical and correlated results about men and women from the paper.
Sample Size -- Line 127: No a priori power analysis presented for sample size of N=11 in each group. Given the relatively small sample size and multiple comparisons, please consider that the sample size needs to be fully justified and increased as needed to ensure rigorous and reproducible findings.
It was unclear to me how the authors calculated the variables such as covered distance, angular deviation, and speed in walking and running. I would suggest you clearly mention the data have been obtained from instruments, and the data processing has been used to exclude the aforementioned variables.

Reviewer 2 ·

Basic reporting

The English language should be grammatically improved to ensure that an international audience can clearly understand the text. Some examples where grammatical or wording errors occur include lines 50 (“Human moving in space has been examined…”), 60 (“at a certain time”), 62 (“human moves”), 72 (“shown that human is able”). Throughout the text, I have highlighted examples in yellow.
Line 71 “or to 20 m (Souman et al., 2009) and eventually return to the same place” is not sufficiently clear. Is this a go and return task?

The sentence “Men, with domination of the left cerebral hemisphere (right-handed, right-legged and right- eyed) turn more to the right than to the left, while women, relying more on the right hemisphere, turn more often to the left than to the right” needs a reference. Moreover, the current phrasing makes comprehension difficult as it incurs that all women rely more on right hemisphere. Whereas hemispheres do work differently in the human brain, gender-brain differences are questioned in neurosciences and recent fMRI data points otherwise. See Nielsen et al. (2013) and follow literature thereafter.

On line 102. The paragraph ends with “These inconveniences can be minimized by practicing orienteering”. The sentence is not clearly linked to the paragraph because it assumes the reader knows which the inconveniences within the many arguments are. I suggest rewriting the paragraph with a direct flow of arguments, in consistent order. Currently, the arguments are ‘going back and forth’.

The research question is well defined and the approach of interest. However, the knowledge gap it aims to cover is not fully clear to the reader. The authors address it in disperse fragments within the introduction. Nevertheless, the reader does not find a cohesive fragment that highlights this point. Since so many of the citations have covered a similar field of knowledge, it is difficult to pin point where this article brings something unique to the field.

Experimental design

The participant consent is not in English. So I cannot review on that. This is ok. I understand translation might be troublesome. But I urge authors to consider any inputs given by reviewers who speak polish.

The researchers should clarify where is Lower Silesia region (country?).

Lines 133-135 present results not methods.

Are “sound-absorbing headphones” the same as noise cancelling headphones (line 157)?

What was the location precision of the GPS device in terms of meters? I understand it has high recording frequency (that is an important report), but its precision is also relevant to this experiment. Moreover, the high frequency was probably used for correction of error points? If so make this available to the reader. i.e. If a point was further than xx it was considered wrong data.

What is the threshold to define when a participant starts moving in a circle? (line 191)

Validity of the findings

I understand that gender had an effect on the results. This is an important aspect which the authors address. However, I miss a stronger argumentation regarding the fact that spatially trained athletes did not differ from non-trained ones. Why isn’t that more highlighted in the discussion? I believe it is an interesting factor. If not explained by lateralization than why would we not be “wired” to move in straight lines? Or is direction of movement very dependent on sensory input/ references? Speculation is welcomed.

Additional comments

Results
Figure 1 is good for the data was transformed to aid interpretation, but the graphics of the image are not at a high standard. The colour code does not facilitate for the discerning between T&F and Foot-O groups. On the legend the sentence “because of stopping the participant before the obstacle” is unclear.

Line 199 to 202 have repeated text?

Table 1 (S1) contains important information but does not follow publication standards. Thereafter it is unnecessarily hard to read through. This visual aspect should be improved in all tables within the article. See as an example:
https://dl.sciencesocieties.org/files/publications/style/chapter-05.pdf

Discussion
Line 277: Considering that no significant differences were observed in the running test. I wonder, since participants where not moving at max speed during walking, why would sexual dimorphism play a role on the results? Or are the authors claiming that anxiety levels differ between genders? And if anxiety differs, why did it not affect the running speed?

The discussion fragment is relevant, but the flow is considerably hard.


Conclusion
Line 365-366: I understand what the authors mean, however the text per-se is unclear. -> Applies to experts when those are deprived of vision and hearing.

Annotated reviews are not available for download in order to protect the identity of reviewers who chose to remain anonymous.

---

## Round 0.2 · Minor Revisions

The revised manuscript has improved considerably and there only a few minor issues that I would like the authors to address before the manuscript can be accepted for publication.

When reporting the results of statistical test, please include the test statistic (e.g. U-value for Mann-Whitney U Test) in addition to the p-value for all statical comparisons. When a significant effect is reported, please also report the median of the dependent variable for both groups/conditions in the text.

It is recommended that data from individual participants are also plotted using scatter plots superimposed over the bar graphs (https://doi.org/10.1371/journal.pbio.1002128). This is particular relevant given the small sample sizes used in the current study. Please add data from additional participants to the bar graphs in figures 2-4.

---

## Round 0.3 · accepted · Accept

The authors have adequately addressed the outstanding comments.